# OpenReview forum: "NeuroGenPoisoning: Neuron-Guided Attacks on Retrieval-Augmented Generation of LLM via Genetic Optimization of External Knowledge"
_NeurIPS.cc/2025/Conference — NeurIPS 2025 poster_

### Official Review · Reviewer_LiGt · 2025-07-01

**Clarity:** 3
**Significance:** 3
**Originality:** 3
**Rating:** 4
**Confidence:** 4

**Summary:**

Retrieval-augmented generation integrates external knowledge during the inference phase of an LLM. However, an adversary can inject poisoned external knowledge to manipulate retrieval content, thereby making an LLM generate malicious outputs. This paper proposes NeuroGenPoisoning, a new attack framework that generates adversarial knowledge in RAG.

**Questions:**

See strengths and weaknesses.

**Ethical Concerns:**

["NO or VERY MINOR ethics concerns only"]

**Final Justification:**

Given that the authors have addressed two out of three comments (as specified in my response to the authors). I kept my score.

**Limitations:**

The limitations of the proposed method can be discussed.

**Paper Formatting Concerns:**

The format looks good.

**Quality:**

3

**Strengths And Weaknesses:**

Strengths:

1. Poisoning attacks on RAG are an important research problem. This paper tries to improve existing attacks, which can help the community to better understand the attacks.

2. The proposed method is evaluated on benchmark datasets and is compared with baselines.

3. The proposed method first finds neurons that are sensitive to contextual information and then optimizes the adversarial external knowledge to maximize the activation of these sensitive neurons. Intuitively, the proposed method makes sense.

Weaknesses:

1. The LLMs used in evaluation are pretty small. The paper may evaluate closed-source LLMs such as GPT-4, Gemini, and so on.

2. The threat model can be clarified. For instance, it is assumed that an attacker has partial access to the model’s internal representations. In general, an attacker either has full access or no access to the model’s internal representations. Could the paper comment on which scenario an attacker has partial access?

3. The LLMs used in evaluation are small in general. The paper may also evaluate whether the proposed method can improve the baseline for closed-source LLMs, e.g., whether the optimized malicious texts can be transferred.

4. Some baselines are missing. For example, GCG can also be extended to optimize the external knowledge to make an LLM generate a target answer. Could the paper comment on the benefit of the proposed method over the GCG baseline? They are briefly discussed in Section 5.1, more discussion would be helpful.

---

> ### Author Rebuttal · Authors · 2025-07-30
>
> We sincerely thank the reviewer for the valuable comments and suggestions. We appreciate the opportunity to clarify and improve our work. We respond accordingly.
>
> **Addressing Weakness 1 and Weakness 3:**
> We appreciate this suggestion. We acknowledge the importance of transferability and evaluation against closed-source LLMs such as GPT-4 and Gemini. We focused our experiments on popular and widely adopted open source LLMs, including LLaMA-2-7B, Vicuna-7B/13B, and Gemma-7B. These models allow internal representation access, which is essential for neuron-level analysis. We will expand our study to some closed-source LLMs to strengthen our claims about attack transferability in our revised version.
>
> **Addressing Weakness 2:**
> We thank the reviewer for pointing this out. We agree that our description of 'partial access' to the internals of the model requires greater precision. In our current setting, the attacker is assumed to have access to neuron activations and gradient-based attribution signals on an LLM, but not to its weights or ability to modify its architecture. This reflects a white-box inference access, commonly adopted in related work such as IRCAN[1] and AutoDAN[2]. To better clarify the scope of access, we provide more detailed explanations as follows.
>
> We consider attacks on an RAG system composed of a frozen LLM $\mathcal{M}$ and a retrieval module $\mathcal{R}$. The adversary cannot modify the model's parameters or training data, and has no access to internal training processes. We assume that the attacker has white-box inference-time access to intermediate neuron activations and can compute attribution signals, such as Integrated Gradients (IG), between inputs and neuron responses. This access allows the attacker to estimate neuron-level Poison-Responsiveness Scores and guide the generation of external knowledge $e^{\text{adv}}$ accordingly. The attacker cannot change model weights or gradients directly, but can interact with the model through controlled forward passes and limited attribution analysis.
>
> **Addressing Weakness 4:**
> We thank the reviewer for this valuable comment. In Section 5.1 and Table 3, we discuss GCG[3] and other recent methods. While GCG uses gradient-based optimization to find adversarial tokens, it does not leverage internal neuron attribution nor is it designed for external knowledge poisoning in RAG. Its effectiveness is mostly limited to short, prompt-based suffixes and lacks scalability for long-form document injection. In contrast, our method explicitly handles knowledge conflicts and adversarial persistence on a scale.
>
> **Reference**
>
> [1] Dan Shi, Renren Jin, Tianhao Shen, Weilong Dong, Xinwei Wu, and Deyi Xiong. IRCAN: Mitigating knowledge conflicts in LLM generation via identifying and reweighting context aware neurons. In The Thirty-eighth Annual Conference on Neural Information Processing Systems, 2024.
>
> [2] Xiaogeng Liu, Nan Xu, Muhao Chen, and Chaowei Xiao. AutoDAN: Generating stealthy jailbreak prompts on aligned large language models. In The Twelfth International Conference on Learning Representations, 2024.
>
> [3] Andy Zou, Zifan Wang, Nicholas Carlini, Milad Nasr, J. Zico Kolter, and Matt Fredrikson. Universal and transferable adversarial attacks on aligned language models. arXiv preprint arXiv:2307.15043, 2023.

---

> > ### Comment · Reviewer_LiGt · 2025-08-06
> >
> > Thanks for the responses. Some of my comments are addressed (except for large LLMs are deferred to the revised version of this paper). I keep my score.

---

### Official Review · Reviewer_PcU3 · 2025-07-02

**Clarity:** 2
**Significance:** 2
**Originality:** 3
**Rating:** 4
**Confidence:** 4

**Summary:**

This paper presents NeuroGenPoisoning, a novel attack framework targeting Retrieval-Augmented Generation (RAG) systems. The authors identify Poison-Responsive Neurons—neurons whose activation is strongly correlated with adversarial external knowledge—and use genetic optimization to evolve poisoned passages that maximize these activations. The method enables large-scale generation of effective adversarial knowledge and is shown to successfully overwrite the model’s internal memory with high success rates across datasets and models. The work also highlights the role of neuron-guided attribution in resolving knowledge conflicts between external and parametric knowledge.

**Questions:**

see weakness.

**Ethical Concerns:**

["NO or VERY MINOR ethics concerns only"]

**Final Justification:**

Thank you for your response. Since most of my concerns have been solved, I will raise my score.

**Limitations:**

Yes

**Quality:**

2

**Strengths And Weaknesses:**

**Strength**:

1.	The target domain of the paper is very interesting; attacking the knowledge component of RAG is a highly promising direction.
2.	The authors validate and visualize their proposed method through comprehensive experiments.

**Weakness**:

1.	The realism of the setting is somewhat limited. While I highly value the problem the authors aim to address and acknowledge the contribution of their work under the current setting, I still have doubts about the practicality of assuming access to model activation values. If this is intended as a white-box attack, I would appreciate a more detailed discussion of this assumption and its applicability in real-world scenarios.
2.	In Line 148, the authors refer to some derivations presented in Appendix 2. However, the equations in Appendix 2 are not numbered and appear to use different LaTeX commands than those used in the main text. I recommend unifying the formatting and ensuring that all equations are properly numbered for clarity and consistency.
3.	In Equation~2, the authors do not seem to provide a mathematical definition for $x$ and $x'$, which could lead to some ambiguity in the formulation of Integrated Gradients (IG). Moreover, the authors do not clarify whether the operation in the equation is a dot product. If it is indeed a dot product, then summing over the input dimensions rather than preserving the per-dimension integration path might be confusing.
4.	In Appendix~2, the authors define $P(e)$ as the sum of the absolute values of the Integrated Gradients (IG). We would like to understand why the sign of IG itself is not considered or discussed. The directionality of IG could carry important semantic meaning, and ignoring the sign might limit interpretability or mask conflicting attribution signals.
5.	At the end of Appendix~2, the authors establish a positive correlation $\Delta \hat{K}_t \propto \Delta P_t$. While this is a clever attempt, it seems that in the equation  \[\frac{d\hat{K}}{dx} = \sum_{(l,i)} \frac{\partial \hat{K}}{\partial f_{l,i}} \cdot \frac{\partial f_{l,i}}{\partial x}\]  the left subterm $\frac{\partial \hat{K}}{\partial f_{l,i}}$ on the right-hand side is not fully addressed in the discussion. A more detailed explanation of this term and its role in the derivation would be helpful.

---

> ### Author Rebuttal · Authors · 2025-07-30
>
> We sincerely thank the reviewer for the valuable comments and suggestions. We appreciate the opportunity to clarify and improve our work. We respond accordingly.
>
> **Addressing Weakness 1:**
> We appreciate the reviewer pointing this out. We agree that the assumption of accessing intermediate activations implies a white-box setting. Our method indeed operates under a white-box setting, assuming access to intermediate activations. While this may not apply to all deployment scenarios, we argue that white-box settings remain practically relevant and important. Moreover, white-box attacks serve a critical role as upper bounds on system vulnerability by offering insights that can inform the design of defenses even in more restricted settings.
>
> **Addressing Weakness 2:**
> We appreciate the reviewer pointing this out. We will revise Appendix B to ensure that all equations are properly numbered and use consistent LaTeX formatting as in the main text. This will enhance readability and maintain clarity throughout.
>
> **Addressing Weakness 3:**
> We thank the reviewer for noting this oversight. We agree that the presentation of the equation in the current version may lead to ambiguity regarding the mathematical operations involved. We provide the following clarifications and improvements:
>
> 1.  **Clarification on the definition of $x$ and $x'$**\
>     In our formulation, x = {q, e} denotes the input composed of both
>     the query $q$ and the candidate external context $e$, and x' = {q}
>     is the corresponding baseline input containing only the query $q$.
>     Both $x$ and $x'$ are tokenized sequences, embedded into the input
>     space prior to attribution computation.
>
> 2.  **Clarification on dot product operation**\
> In the equation in Section 3.2,
>    $$IG_{(l,i)} = \int_{\alpha=0}^{1} \frac{\partial f_{l,i}\left(x' + \alpha(x - x')\right)}{\partial x} \cdot (x - x') \ d\alpha$$
> the operation $\frac{\partial f_{i,i}}{\partial x} \cdot (x - x')$
>     in the IG expression is indeed a dot product, following the standard
>     Integrated Gradients (IG) definition by Sundararajan et al.\[1\].
>     This yields a scalar attribution score per neuron $f_{i,i}$ that
>     reflects the total contribution of external knowledge perturbation
>     along the attribution path.
>
> 3.  **Clarification on summing and dimension preservation**\
>     While it is possible to preserve attribution along each input
>     dimension (i.e., avoid summation), we adopt the dot product form to
>     produce a scalar relevance score per neuron for purposes of ranking
>     and selecting top-r neurons. This design choice is consistent with
>     our goal to aggregate attribution across input tokens and simplify
>     neuron-level optimization. we acknowledge that the reviewer's
>     suggestion to consider the per-dimension integration path highlights
>     a useful perspective, and it is helpful to identify a clearer way to
>     express our formulation. In light of this, we propose the following
>     improvement to Equation (1) for clarity and mathematical rigor:
>     $$IG_{(l,i)}(x) = (x - x')^\top \cdot \int_{\alpha=0}^1 \frac{\partial f_{l,i}(x' + \alpha(x - x'))}{\partial x} \ d\alpha$$
>
> **Addressing Weakness 4:** We appreciate the reviewer's insightful
> observation. We agree that the sign of Integrated Gradients (IG) can
> carry meaningful interpretability signals, especially in tasks where
> distinguishing positive versus negative contributions to neuron
> activation is important. In our approach, we chose to use the absolute
> values of the IG to quantify overall attribution strength, agnostic to
> positive or negative influence. We define $\mathcal{P}(e)$ as the sum of
> Integrated Gradients (IG) attribution over a global set of top-$r$
> neurons $\mathcal{N}_{\text{top}-r}$:
>
> $P(e) = \sum_{(l,i) \in N_{\text{top-}r}} |IG_{(l,i)}|$
>
> This scalar quantity enables a consistent ranking of external knowledge
> $e$ for neuron optimization across inputs, which simplifies the
> aggregation.
>
> **Addressing Weakness 5:**
> We appreciate the reviewer's insightful observation. We agree that $\frac{\partial \hat{K}}{\partial f_{i,i}}$ plays a significant role in the chain-rule decomposition of the gradient $\frac{d\hat{K}}{dx}$. In our framework, $\hat{K}$ denotes the predicted performance proxy under a poisoned configuration. The term $\frac{\partial \hat{K}}{\partial f_{i,i}}$ quantifies how sensitive the final model performance is to changes in the activation of neuron $(l, i)$. The reviewer's valuable comments are helpful in revising Appendix B. We provide more detailed explanations as follows.
> $$
> \frac{d\hat{K}}{dx} = \sum_{(l,i)} \frac{\partial \hat{K}}{\partial f_{l,i}} \cdot \frac{\partial f_{l,i}}{\partial x}
> $$
>
> (1) $\hat{K}$ represents a scalar performance proxy that is a function of the entire network computation; (2) $f_{l,i}$ denotes the activation of the neuron $i$ in the layer $l$, which is a function of the input $x$; (3) $\frac{d\hat{K}}{dx}$ is the total derivative that captures how perturbations in the input $x$ affect the final output of the model.
>
> By applying the chain rule, the derivative is decomposed into two multiplicative components:
>
> (1) $\frac{\partial f_{l,i}}{\partial x}$: This term captures the level of tolerance of the activation of a neuron to changes in input $x$. This is what Integrated Gradients (IG) estimates;
>
> (2) $\frac{\partial \hat{K}}{\partial f_{l,i}}$: This term measures how a neuron $f_{l,i}$'s activation affects the final output of models $\hat{K}$. It emphasizes the importance of that neuron for the model's prediction.
>
> **Reference**
>
> [1] Mukund Sundararajan, Ankur Taly, and Qiqi Yan. Axiomatic attribution for deep networks. In Doina Precup and Yee Whye Teh, editors, Proceedings of the 34th International Conference on Machine Learning, volume 70 of Proceedings of Machine Learning Research, pages 3319–3328. PMLR, 06–11 Aug 2017.

---

> > ### Comment · Reviewer_PcU3 · 2025-08-08
> >
> > Thank you for your response. Since most of my concerns have been solved, I will raise my score.

---

### Official Review · Reviewer_QhjU · 2025-07-03

**Clarity:** 3
**Significance:** 3
**Originality:** 3
**Rating:** 5
**Confidence:** 4

**Summary:**

This paper proposes a novel white-box attack that poisons RAG by optimizing external passages to activate specific neurons in large language models. The method uses Integrated Gradients to identify the "Poison-Responsive Neurons" and evolves adversarial passages through a genetic algorithm that maximizes their activation. Experiments across multiple models and QA datasets show that the attack can reliably override the model’s internal knowledge, achieving a success rate of over 90% in generating incorrect answers, highlighting a new vulnerability in LLMs and raising concerns about how external knowledge can be manipulated to subvert trusted outputs.

**Questions:**

1. Do you have other evaluations on Poison-Responsive Neurons with different model architectures, scales, or alignment methods (e.g., RLHF)?
2. What’s the potential approaches or research directions to extend your methodology to black-box scenarios?
3. How does your method compare to other white-box attack methods, such as GCG or AutoDAN?

**Ethical Concerns:**

["NO or VERY MINOR ethics concerns only"]

**Final Justification:**

This paper proposes a novel framework for attacking RAG. Considering its contribution and demonstrated effectiveness, and given that most of my concerns have been addressed, I rate this paper as an accept.

**Limitations:**

Yes.

**Quality:**

3

**Strengths And Weaknesses:**

Strengths:

1. The paper addresses a critical but underexplored issue in RAG security.
2. It introduces a novel white-box attack framework that leverages neuron attribution and genetic search to directly manipulate model behavior.
3. The method achieves high effectiveness and robustness, with over 90% POSR across multiple QA tasks and open-source LLMs.
4. The concept of knowledge conflict provides an effective theoretical framework for understanding the internal model vulnerabilities.

Weaknesses:
1. The proposed attack relies on white-box access to internal gradients, limiting its practicality for real-world black-box LLMs (although acknowledged by the authors).
2. The evaluation only compares to PoisonedRAG while missing other relevant white-box and genetic attack baselines.
3. The claim of universal "Poison-Responsive Neurons" lacks cross-architecture validation, leaving their generality and stability unproven.

---

> ### Author Rebuttal · Authors · 2025-07-30
>
> We sincerely thank the reviewer for the valuable comments and suggestions. We appreciate the opportunity to clarify and improve our work. We respond accordingly.
>
> **Addressing Weakness 1 \& Question 2:**
> We appreciate the reviewer highlighting this. As we mentioned in our paper, our method currently assumes white-box access to compute neuron-level attribution via Integrated Gradients. This setting allows the attacker to estimate neuron-level Poison-Responsiveness Scores and guide the generation of external knowledge $e^{\text{adv}}$ accordingly. Our design was intentional in studying the internal dynamics of poisoning under RAG and in identifying the causal roles of specific neurons. We agree that real-world black-box LLM scenarios are highly relevant. We mentioned our limitation and the future work direction in Section 6. To be more specific, it is crucial to find ways to estimate the neuron sensitivity without requiring gradient access. Without gradients, it is possible to use reinforcement learning, where the reward is based on the observed success of the overriding model output.
>
> **Addressing Weakness 2 \& Question 3:**
> We thank the reviewer for this important point. Considering the consistency of training and attacking goals, we compare primarily with PoisonedRAG. We agree that a broader comparison is necessary to emphasize our contributions. We also provided comparisons with other related methods in Section 5.1 and Table 3. We provide some details as follows. We will include this further discussion and our detailed results in the revised version.
>
> (1) AutoDAN[1]: as we discussed in Section 5.1, AutoDAN does not target internal conflict or model memory override. In addition, AutoDAN lacks neuron guidance and is designed for prompt-suffix attacks, not knowledge-based poisoning.
>
> (2) GCG[2]: GCG performs token-level prompt attacks based on gradients. It does not optimize document-scale knowledge or account for neuron-level conflict.
>
> **Addressing Weakness 3 \& Question 1:**
> We thank the reviewer for raising this important point regarding Poison-Responsive Neurons. We clarify and respond as follows. In our main experiments (shown in Section 4.4, Table 1, Figure 1 and Figure 2), we include evaluations on 4 different LLMs with varying sizes and alignment methods. We observed that the selected top-$r$ Poison-Responsive Neurons consistently maintained strong activation patterns and guided effective poisoning across all models ($>$ 90\% final POSR).
>
> **Reference**
>
> [1] Xiaogeng Liu, Nan Xu, Muhao Chen, and Chaowei Xiao. AutoDAN: Generating stealthy jailbreak prompts on aligned large language models. In The Twelfth International Conference on Learning Representations, 2024.
>
> [2] Andy Zou, Zifan Wang, Nicholas Carlini, Milad Nasr, J. Zico Kolter, and Matt Fredrikson. Universal and transferable adversarial attacks on aligned language models. arXiv preprint arXiv:2307.15043, 2023.

---

> > ### Comment · Reviewer_QhjU · 2025-08-06
> >
> > Thank you for your response, which partly addressed my concerns. I will keep my score.

---

### Official Review · Reviewer_i5DG · 2025-07-03

**Clarity:** 3
**Significance:** 3
**Originality:** 3
**Rating:** 4
**Confidence:** 3

**Summary:**

This paper proposes NeuroGenPoisoning, a framework for generating adversarial attacks against Retrieval-Augmented Generation (RAG) systems. The method focuses on its use of internal model signals to guide the creation of poisoned external knowledge. Specifically, it identifies "Poison-Responsive Neurons"—internal units that are sensitive to external context—using Integrated Gradients. It then employs a genetic algorithm to evolve adversarial text passages that maximize the activation of these target neurons, with the goal of overriding the LLM's correct parametric knowledge and forcing it to output an attacker-chosen answer.
The authors evaluate their method on several QA datasets and LLMs, reporting high success rates (over 90% Population Overwrite Success Rate) in forcing the model to adopt the poisoned knowledge, even in cases of strong "knowledge conflict" where the model's internal memory is correct.

**Questions:**

1. Can you provide a stronger justification for the "global neuron set" assumption? Have you performed any analysis to test the generalization of this set across different knowledge domains or on out-of-distribution queries?
2. Given the severe limitations of the white-box model, what are the concrete, non-trivial steps required to approximate this attack in a more realistic, query-based black-box setting? Is there any preliminary evidence to suggest this is feasible?
3. Could you provide an analysis of the computational cost of NeuroGenPoisoning? For instance, how many forward/backward passes are required on average to generate a successful attack for a single query, and how does this compare to simpler baselines?
4. How would your method's performance and the "stealthiness" of its outputs change if the genetic algorithm's fitness function included a penalty for stylistic deviation from a target corpus (e.g., Wikipedia), in addition to maximizing neuron activation?

**Ethical Concerns:**

["NO or VERY MINOR ethics concerns only"]

**Quality:**

3

**Strengths And Weaknesses:**

Strengths:
1. In my opinion, the paper introduces an interesting conceptual direction for adversarial attacks by shifting the optimization target from surface-level text features or output probabilities to the activation of specific internal neurons. This white-box approach provides a more principled way to explore the causal mechanisms of context overriding in LLMs.
2. The work's explicit focus on the "knowledge conflict" scenario is commendable. This is a particularly challenging setting for RAG poisoning, and demonstrating an ability to override strongly held factual beliefs is an important stress test for model robustness.
3. The use of a genetic algorithm is a suitable choice for this problem, as it allows for the exploration of a diverse semantic space of adversarial content, potentially leading to more fluent and varied attacks than token-level gradient optimization might produce.

Weaknesses:
1. The most significant weakness of this work is its reliance on a pure white-box threat model. The method requires full access to the model's internal states, including the ability to compute gradients (for IG) and read neuron activations for every candidate in the genetic algorithm's population. This is highly unrealistic for almost all deployed RAG systems, which are typically accessed via APIs. This constraint severely limits the practical relevance of the attack and frames the contribution more as a theoretical exploration than a demonstration of a real-world vulnerability.
2. The core mechanism relies on the assumption that a small, fixed set of "Poison-Responsive Neurons" (e.g., r=10) identified from a seed set can be used to effectively guide attacks across all queries. This is a strong claim with insufficient evidence. The paper does not analyze whether this neuron set generalizes across different knowledge domains (e.g., history vs. science) or question types. It is highly plausible that different neural circuits are responsible for different reasoning tasks, which would invalidate this "global" set assumption and undermine the method's robustness.
3. The paper's claim of generating "stealthy" poisoned knowledge is not convincingly supported. The sole quantitative metric is Relative PPL Drop, which is a poor proxy for human-perceived fluency or stylistic appropriateness. The examples in the appendix often contain tell-tale artifacts (e.g., "Authoritative sources indicate that...", "According to a recent study...") that make them look artificial and unlike genuine retrieved documents. A proper evaluation of stealthiness would require human studies or more sophisticated stylistic analysis.
4. The paper emphasizes the ability to generate a large volume of successful attacks but completely ignores the associated computational cost. A process involving a genetic algorithm that requires repeated forward and backward passes through a large language model for every candidate in every generation is extremely resource-intensive. This makes the attack not only impractical from an access perspective but also from a computational-cost perspective, further limiting its real-world feasibility.

---

> ### Author Rebuttal · Authors · 2025-07-30
>
> We sincerely thank the reviewer for the valuable comments and suggestions. We appreciate the opportunity to clarify and improve our work. We respond accordingly.
>
> **Addressing Weakness 1 \& Question 2:**
> We appreciate the concern of the reviewer about the white-box assumption. As we mentioned in our paper, our method currently assumes white-box access to compute neuron-level attribution via Integrated Gradients. This setting allows the attacker to estimate neuron-level Poison-Responsiveness Scores and guide the generation of external knowledge $e^{\text{adv}}$ accordingly. Our design was intentional in studying the internal dynamics of poisoning under RAG and in identifying the causal roles of specific neurons. We agree that real-world black-box LLM scenarios are highly relevant.
>
> **Addressing Weakness 2 \& Question 1:**
> We thank the reviewer for pointing out the need for deeper analysis in different knowledge domains and query types. To clarify, our evaluation already covers a wide range of domains, as we conduct experiments on three benchmark open-domain QA datasets, SQuAD 2.0, TriviaQA, and WikiQA. Each contains questions from diverse topics such as history, science, technology, sports, popular culture, and so on. We conducted a stratified analysis on the three datasets, grouping queries into the following knowledge domains: history \& geography, literature, science \& technology, and popular culture. For each domain, we measured Population Overwrite Success Rate (POSR) using the same global top Poison-Responsive Neurons set. We observe that POSR remains high (above 90\%) across all datasets and models, indicating that the attack is not limited to any specific domain or query structure. The following table shows the detailed results for different domains across three datasets of knowledge.
>
> | Knowledge Domains     | LLaMA-2-7B | Vicuna-7B | Vicuna-13B | Gemma-7B |
> |-----------------------|------------|-----------|-------------|----------|
> | history & geography   | 0.92       | 0.91      | 0.93        | 0.94     |
> | literature            | 0.93       | 0.94      | 0.94        | 0.92     |
> | science & technology  | 0.90       | 0.92      | 0.92        | 0.91     |
> | popular culture       | 0.94       | 0.90      | 0.95        | 0.93     |
>
> **Addressing Weakness 3 \& Question 4:**
> We appreciate the concern of the reviewer about stealthiness. As the reviewer correctly points out, some generated passages exhibit overly templated or formulaic expressions. We agree that future versions of our work would benefit from human evaluation or retrieved style scoring. We will expand on this work in our revised version.
>
> **Addressing Weakness 4 \& Question 3:**
> We thank the reviewer for this important point about computational cost. We acknowledge that the genetic algorithm is a resource-intensive process. We provide the computational cost analysis as follows.
>
> If we set population size to 100, the number of generations to 10, the total number of candidates per query to 1,000, each candidate requires 1 forward pass for the model response and 1 forward pass and $n$ backward passes to compute Integrated Gradients (IG). In this setting, for a single query, the total computational cost is $(1000+1000n)$ forward passes and $1000n$ backward passes. We will include this detailed computational cost analysis in our revised version. We acknowledge that computational cost is a significant metric for a novel attack method of LLMs. Despite the cost, our proposed method achieves a balance between effectiveness, interpretability, and scalability not matched by existing methods.
>
> **Reference**
>
> [1] Mukund Sundararajan, Ankur Taly, and Qiqi Yan. Axiomatic attribution for deep networks. In Doina Precup and Yee Whye Teh, editors, Proceedings of the 34th International Conference on Machine Learning, volume 70 of Proceedings of Machine Learning Research, pages 3319–3328. PMLR, 06–11 Aug 2017.

---

> > ### Comment · Reviewer_i5DG · 2025-08-08
> >
> > Thanks for the responses. Some of my comments are addressed (except for W3 about the stealthiness). I keep my score.

---

### Note · Authors · 2025-08-12

We thank all the reviewers and AC for their efforts during the review and discussion period, and we appreciate all the valuable feedback. We are pleased that all reviewers recognized our strengths and main contributions.

We provided detailed explanations to the reviewers' concerns in the rebuttal period. We have clarified the issues with our white-box problem setting and have explained why we did it this way and what the practical implications are. We also addressed the concerns for the baselines, generalization, evaluation, and computational cost. We provided a detailed comparative analysis of other attack methods and different knowledge domains. Meanwhile, detailed explanations and appropriate revisions have been provided for formulas and Mathematical problems in the main text and the Appendix. Regarding the concern about larger LLMs, as discussed in our rebuttal, further exploration in closed-source LLMs could strengthen the generalization and transferability.

In summary, we introduce a novel RAG poisoning attack framework in our paper, which combines neuron-level attribution with genetic optimization to craft adversarial external knowledge in RAG systems at scale. We believe this represents a meaningful step forward for LLM security, and we appreciate the opportunity for consideration.

---

### Decision · Program_Chairs · 2025-09-17

**Decision:**

Accept (poster)

**Comment:**

**Summary**

This paper introduces a novel attack method against RAG systems by analyzing the internal neuron activations of the underlying LLM. The core hypothesis is that a small, identifiable subset of neurons is particularly susceptible to poisoning. Leveraging this, the authors employ a genetic algorithm to optimize the content of poisoned passages based on the activation of these target neurons.

**Strengths**

* The paper introduces a new attack for RAG systems rooted in the internal mechanics of LLMs. This is an interesting direction that could inspire future work in both adversarial attacks and defenses.
* The work connects the proposed attack to the existing literature on knowledge conflicts in LLMs, offering valuable insights for the development of more robust and reliable RAG systems.

**Weaknesses**

The reviewers reached a clear consensus on several major weaknesses:

* **Practicality of the Threat Model:** The proposed attack relies on a white-box assumption, requiring full access to the target model's internal states. This significantly limits its applicability, as many real-world RAG systems are deployed using closed-source LLMs. The authors acknowledged this limitation but did not offer a path toward a more realistic threat model.
* **Unsupported Core Assumption:** The central claim that, a universal and identifiable set of "poison-responsive" neurons exists, lacks sufficient empirical or theoretical justification. The evidence provided is preliminary, and this foundational hypothesis remains insufficiently validated.
* **Insufficient Baselines:** The experimental comparison against existing attack methods is not comprehensive enough to robustly contextualize or demonstrate the relative effectiveness of the proposed technique.

Beyond the points raised by the reviewers, there is a significant high-level omission:

* **Missing Discussion on Defenses:** The goal of attack research is ultimately to strengthen system security, yet this paper fails to explore the implications for defenders. Given the stated connection to knowledge conflicts, a crucial and obvious question is whether existing methods for mitigating such conflicts could also serve as a defense against this attack. This omission limits the paper's contribution to the broader community.

**Recommendation**

The reviewers, while in general borderline positive, have correctly highlighted significant weaknesses, particularly the restricted white-box setting and insufficient evidence for the paper's core hypothesis. These are serious flaws that limit the work's immediate impact.

I lean towards acceptance because the central idea of targeting "poison-sensitive" neurons could inspire interesting follow-up research into model vulnerabilities. That said, I would not be surprised if the program committee ultimately decides to reject as the paper's limitations are too significant in its current form.